

# The viscosity and surface tension of supercooled levitated droplets determined by excitation of shape oscillations

Mohit Singh[†1], Stephanie H. Jones[†1], Alexei Kiselev[1], Denis Duft[1], Thomas Leisner[1,2]

[1]Karlsruhe Institute of Technology, Institute of Meteorology and Climate Research, Karlsruhe, 76021, Germany
[2]University of Heidelberg, Institute of Environmental Physics, Heidelberg, 69120, Germany
†both authors contributed equally to this work

*Correspondence to*: Denis Duft (denis.duft@kit.edu)

**Abstract.** We report a new method for determining the viscosity and surface tension of supercooled liquid droplets using electrodynamic levitation and phase analysis of shape oscillations. The method uses a high frequency alternating electrical
potential to excite shape oscillations in a levitated droplet, and the phase shift of the oscillations is used to simultaneously determine droplet viscosity and surface tension. The advantages over existing contactless methods include its applicability to atmospherically relevant temperatures, and the possibility to continuously monitor changes in real time. We demonstrate proof-of-concept measurement for supercooled water droplets and dilute sucrose solution droplets, and we anticipate that the technique could be used to measure viscosity values within the semi-solid range for droplets containing dilute organics. The
technique is especially well-suited for investigation of the role of atmospheric processing on the viscosity and surface tension of solution droplets in equilibrium with a given or changing relative humidity.

## 1 Introduction

Atmospheric aerosols impact the radiative budget of the climate, air quality and human health (Masson-Delmotte et al., 2021; Shiraiwa et al., 2017). The physicochemical and optical properties of aerosols influence how aerosols scatter and absorb solar
radiation and also dictate their ability to form Cloud Condensation Nuclei (CCN) and Ice Nuclei (IN) (Masson-Delmotte et al., 2021). Viscosity and surface tension are important physical properties of aerosol which influence numerous atmospheric processes. Aerosol viscosity can impact: the rate of heterogeneous and photochemical reactions; lifetimes of chemical species; evaporation, and growth processes leading to CCN formation, as well as deposition and the ability to act as an IN (Reid et al., 2018). Additionally, aerosol morphology and phase state are connected to viscosity (Koop et al., 2011; Zobrist et al., 2008)
and knowledge of aerosol surface tension informs on cloud droplet formation (e.g. Ovadnevaite et al., 2017). Owing to the complex chemical composition of atmospheric aerosol, the dynamic nature of the atmosphere, and the ability of aerosols to undergo hygroscopic growth; viscosity and surface tension are likely to evolve during an aerosol lifetime. Therefore, in order to better understand the impact of atmospheric aerosol on the climate, it is necessary to characterize aerosol properties such as viscosity and surface tension over a range of atmospherically relevant conditions such as temperature and Relative Humidity
(RH).



To date, numerous viscosity measurements have been conducted on bulk material and smaller sample volumes such as particles (e.g. Reid et al., 2018). A variety of contact and non-contact methods have been used for smaller sample volumes including: bounce measurements on an impactor (Virtanen et al., 2010), poke-flow (Murray et al., 2012; Renbaum-Wolff et al., 2013a), bead mobility (Renbaum-Wolff et al., 2013b) and Atomic Force Microscopy (AFM) measurements (Qin et al., 2021). Given

the natural environment of an aerosol, and the limited sample volume obtained in field studies, non-contact methods that use low sample volumes, such as those achieved using single particle levitation provide the optimum environment in which to study aerosol viscosity. Furthermore, such techniques also ensure no measurement artefact from contact with a substrate.

Measurements made using non-contact methods have focused on determination of viscosity and/or surface tension of levitated droplets at ambient temperatures (Athanasiadis et al., 2016; Bzdek et al., 2015; Bzdek et al., 2020; Endo et al., 2018; Fitzgerald

et al., 2016; Power and Reid, 2014; Power et al., 2013; Rafferty et al., 2019; Richards et al., 2020; Tong et al., 2022). Several studies have probed the collision of two levitated droplets (optically or electrodynamically levitated) to determine viscosity and surface tension from the relaxation time and oscillation frequency, respectively following collision (Bzdek et al., 2015; Bzdek et al., 2020; Power and Reid, 2014; Power et al., 2013; Richards et al., 2020; Tong et al., 2022). (Power and Reid, 2014; Power et al., 2013) determined viscosities over the range $10^{-3}$ to $10^9$ Pa s for droplets of volume < 500 femtolitres for single,

binary and ternary component droplets whereas Bzdek and co-workers determined values of surface tension and viscosity for picolitre droplets containing glutaric acid and sodium chloride and glutaric acid droplets doped with surfactant (Bzdek et al., 2015; Bzdek et al., 2020). Richards et al. (2020) studied gel formation in levitated aerosol with a dual electrodynamic balance based on a linear quadrupole design with a second counterbalance. Viscosity values in the range $10^4$ Pa s were determined for sorbitol and glucose, using the characteristic time for relaxation, and surface tension data was extrapolated from the literature.

Fluorescence lifetime imaging has also been used to determine viscosity in optically levitated droplets at ambient temperature (Fitzgerald et al., 2016; Athanasiadis et al., 2016). The fluorescence lifetime of molecular rotors added to a levitated droplet is used to determine viscosity. Fitzgerald et al. (2016) explored the effect of relative humidity on viscosity for sucrose and pharmaceutically relevant NaCl and salbutamol sulfate aerosol for droplets in the size range 2-12 μm. The authors cautioned that for fluorescent lifetimes of > 2 ns, the correlation between viscosity and fluorescence lifetime becomes non-linear and

viscosity values obtained at longer fluorescent lifetimes should be taken as an order of magnitude approximation.

Endo et al. (2018) determined the surface tension at ambient temperature of optically levitated droplets of ammonium sulfate and mixed ammonium sulfate and sodium dodecyl sulfate droplets. The authors assumed that a spherical spontaneous resonance arose from a thermally induced capillary wave and obtained droplet surface tension from analysis of the light scattering signal. Surface tension was determined with a precision of ±1 mN m$^{-1}$ on droplets in the size range 4.7 μm to

12.4 μm radius.

Limited non-contact viscosity measurements have been performed at atmospherically relevant lower temperatures. Järvinen et al. (2016) conducted a chamber experiment on α-pinene secondary organic aerosol and monitored the depolarization in backscattered light intensity to determine particle asphericity and thus estimate viscosity at temperatures of -10, -20, -30 and -38°C. Relative humidity was gradually increased throughout these experiments to determine the RH range where a viscosity



transition occurred i.e. a phase change from semi-solid to liquid indicated by an abrupt change to a spherical shape as detected by depolarization measurements. Rothfuss and Petters (2017, 2016) studied nanoparticle dimer formation from the coagulation of charged monomers in a chamber over a range of temperatures and RHs. Mobility diameter measurements were conducted to assess the sintering rate of dimers and determine aerosol phase state over a wide RH range for atmospherically relevant organic material and inorganic salts (Rothfuss and Petters, 2016). Further measurements of viscosity and surface tension for

sucrose nanoparticle dimers were also made as a function of temperature and RH. These studies explored a temperature range spanning from of -11°C to 80°C.

Trinh and Ohsaka (1995) measured a number of physical properties of acoustically levitated supercooled droplets. The amplitude of the acoustic force was modulated to induce shape oscillations in millimetre size levitated droplets from which the surface tension and viscosity were obtained. Surface tension was determined by measurement of the resonance frequency

of the induced shape oscillations and viscosity was determined from the decay time after the force modulation was stopped. Measurements were made for water droplets in the mm size range down to -21°C. Large oscillations, i.e. on the order of 1-2 % in droplet diameter, were induced and analysed for surface tension. Rafferty et al. (2019) also studied deformations induced by laser power in optically tweezed droplets to determine surface tension values from changes in cavity enhanced Raman scattering.

Previously, shape oscillations of highly charged glycol and water droplets levitated in an Electrodynamic Balance (EDB) were studied (Duft et al., 2002; Giglio et al., 2008). The shape oscillations were driven by the electric field applied for levitation. Duft et al. (2002) measured the phase and amplitude of oscillation to validate the Rayleigh instability criteria. It was further proposed that analysis of such shape oscillations could be used to determine the viscosity and surface tension of levitated droplets.

To summarise, to date there is no generally accepted single technique capable of determining the different ranges of viscosity and surface tension over a broad range of environmentally relevant conditions e.g. temperature and relative humidity. Non-contact methods are ideally suited to measurement of aerosol because of the use of small sample volumes enabling the study of real atmospheric samples and no measurement artefact from a substrate. Although a wide range of viscosities can be experimentally determined from relaxation following collision of levitated droplets, the analysis requires different methods

depending on the viscosity range i.e. viscosity up to 10 Pa s: light scattering, viscosity > $10^7$ Pa s: brightfield imaging and viscosity > $10^7$ Pa s: reappearance of whispering gallery modes in cavity enhanced Raman spectra (Power and Reid, 2014; Power et al., 2013). Furthermore, the majority of viscosity studies have been performed at ambient temperatures with concentrated organic solutions, meaning that measurements at lower temperatures and on more dilute aerosol droplets formed through hygroscopic growth, are lacking. Such measurements are important given the potential for fast aqueous chemistry and

photochemistry to occur and to determine the effect of viscosity on reaction kinetics. Currently, viscosity values for dilute droplets are derived from models.

Here we report a novel method to simultaneously measure the temperature dependent viscosity and surface tension of charged droplets levitated in an electrodynamic balance. In addition to the alternating electric field required for levitation, a secondary



electric field of variable frequency is applied to induce shape oscillations in the levitated droplet. The shape oscillations are

analysed by light scattering and the phase shift of the induced shape oscillations with respect to the second electric field is then used to determine droplet viscosity and surface tension. Here, we report the proof-of-concept measurements performed on water droplets and sucrose solution droplets over the atmospherically relevant temperature range between ~0°C to -31°C.

## 2 Method

### 2.1 Theoretical background

The equilibrium shape of a free drop is spherical because it represents the configuration with the lowest surface energy for the system. When a drop is perturbed from equilibrium, the inertia of the system tends to drive the drop away from equilibrium and surface tension acts to restore the drop shape resulting in shape oscillations. Lord Rayleigh (1882) studied the shape oscillations of an inviscid and perfectly conducting free droplet surrounded by an insulating gas under the influence of inherent charge and surface tension. He expanded the droplet shape into an infinite series of spherical harmonic functions $Y_{lm}(\Theta, \phi)$

and derived the frequency of oscillations of its terms. Hasse (1975) later generalized this approach for the case of a charged viscous drop. He could show that the equation of motion for each mode $l$ is given by

$$M_l \ddot{\alpha}_l + Z_l \dot{\alpha}_l + C_l \alpha_l = F_{ext,l} \tag{1}$$

where, $\alpha_l$ is the deformation coefficient of the corresponding $l^{th}$ term of the spherical harmonics expansion. In Eq. (1) we added the term $F_{ext,l}$ to account for an external driving force exciting mode $l$. The coefficients are given as

$$M_l = \frac{\rho R^5}{l} \tag{1a}$$

$$Z_l = 2\eta R^3 (2l+1)(l-1)/l \tag{1b}$$

$$C_l = \left(\sigma R^2 (l+2) - \frac{Q^2}{(16\pi^2 \varepsilon_0 R)}\right)(l-1) \tag{1c}$$

where, $\varepsilon_0$ is the electrical permittivity of free space, $R$ and $Q$ are radius und charge of the droplet, and $\rho$, $\sigma$ and $\eta$ are the density, the surface tension and the viscosity of the droplet, respectively.

At this point we will make use of the fact that the alternating quadrupole potential of the EDB is in the form of $Y_{2,0}$ and preferentially excites the corresponding quadrupole ($l=2$, $m=0$) mode of droplet shape oscillation. The external forcing on the quadrupole mode of oscillation will have two components, one as a result of the low frequency trapping potential at angular frequency $\omega_t$, and the other from the high frequency excitation potential at angular frequency $\omega_e$. The equation of motion for the quadrupole shape oscillation in the presence of the levitation and excitation potential is:

$$M_2 \ddot{\alpha}_2 + Z_2 \dot{\alpha}_2 + C_2 \alpha_2 - F_t \sin(\omega_t t) - F_e \sin(\omega_e t) = 0 \tag{2}$$

where we included the driving forces due the low frequency trapping ($F_t$) and high frequency excitation electric fields ($F_e$). Equation (2) can be rewritten to obtain:

$$\ddot{\alpha}_2 + 2\gamma \dot{\alpha}_2 + \omega_0^2 \alpha_2 - \xi_t \sin(\omega_t t) - \xi_e \sin(\omega_e t) = 0 \tag{3}$$



This is the equation of a damped harmonic oscillation with harmonic excitation, where we identify $\gamma$ as the damping constant and $\omega_0$ as the natural angular frequency of the droplet shape oscillation with the coefficients:

$$\gamma = \frac{Z_2}{2M_2} \tag{3a}$$

$$\xi_T = F_t/M_2 \text{ and } \xi_e = F_e/M_2 \tag{3b}$$

$$\omega_0^2 = \frac{C_2}{M_2} \tag{3c}$$

Since Eq. (3) is linear with respect to $\alpha$, the droplet's response to the combined influence of two driving fields will be the sum of their two independent responses. The steady state solution of Eq. (3) can be written as:

$$\alpha_2(t) = A_t \sin(\omega_t t - \phi_t) + A_e \sin(\omega_e t - \phi_e) \tag{4}$$

where, $A_t$ and $A_e$ are the amplitudes of oscillation and $\phi_t$ and $\phi_e$ are the phase shifts with respect to the phase of the external driving forces. After inserting Eq. (4) into Eq. (3) and collecting the corresponding sine and cosine coefficients, the phase shift with respect to the excitation frequency is found to be:

$$\phi_e = \tan^{-1} \frac{2\gamma \omega_e}{(\omega_e^2 - \omega_0^2)} \tag{5}$$

If the phase shift between the driving force and response is measured over a sufficiently wide frequency interval, both $\gamma$ and $\omega_0$ can be obtained experimentally by fitting Eq. (5). Once these parameters are known, the viscosity and surface tension of the droplet can be calculated independently using

$$\eta = \frac{\rho R^2 \gamma}{5} \tag{6}$$

and

$$\sigma = \frac{1}{4R^2} \left( \rho R^5 \omega_0^2 + \frac{Q^2}{16 \pi^2 \varepsilon_0 R} \right) \tag{7}$$

Equations (6) and (7) follow from inserting (3a) and (3c) into (1a-c) and solving for $\eta$ and $\sigma$. It is important to emphasize that in this analysis, the amplitudes of both, the droplet oscillation and the driving force, are not required. From Eq. (6) and (7) it becomes clear that in addition to $\gamma$ and $\omega_0$, the parameters $\rho$, $Q$ and $R$ have to be known. The procedures to determine the latter quantities are detailed in the Supplementary Information (SI).

### 2.1 Experimental setup and procedure

Charged droplets are levitated in a quadrupole EDB. A detailed description of the EDB has been reported in a previous publication (Rzesanke et al., 2012) and Sect. 1 in the SI provides further information. An overview of the experimental setup is given in Figure 1 and will be briefly described here. The EDB is of classical hyperboloidal design (Paul, 1990) in order to provide $l$=2 excitation. The trapping Alternating Current (AC) potential at frequency $\omega_t$ is applied to the central electrode. The potential applied to the endcap electrodes consists of the excitation AC potential at frequency $\omega_e$ overlaid with a Direct Current (DC) potential which compensates the gravitational force on the droplet. The EDB is mounted in a temperature-controlled and thermally insulated vacuum housing. Individual charged droplets are generated using a piezo-driven nozzle and a HeNe



(632.8nm) laser is directed onto the levitated droplet. One part of the scattered light is guided to the Photomultiplier Tube
(PMT) (Hamamatsu H10723-20) while another fraction is used to obtain angular resolved light scattering images using a
Charge-Coupled Device (CCD) camera. The vertical position of the droplet in the trap is controlled by a feedback loop that
adjusts the DC potential applied to the endcap electrodes based on position information obtained from a signal on a CCD line
detector. A custom-built High Frequency (HF) high voltage AC generator is used to excite droplet shape oscillations. For a
more detailed description of the HF generator see Sect. 2 of the SI.

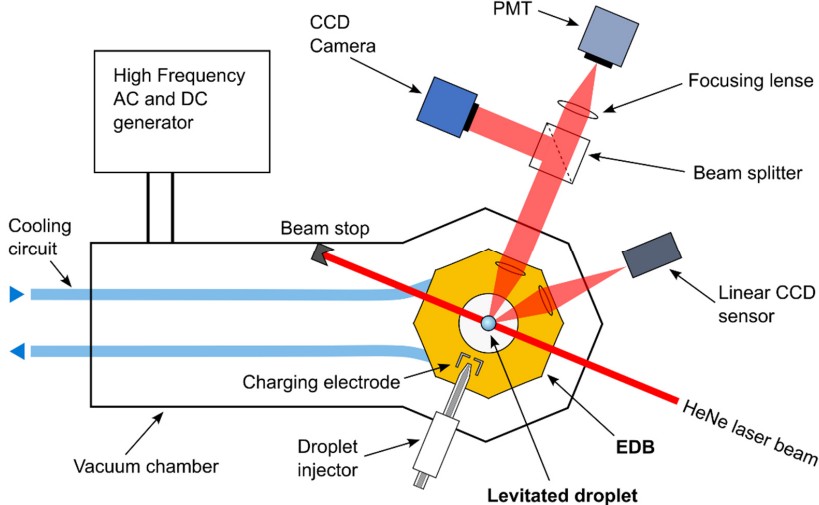


**Figure 1 Schematic of the experimental setup depicted from an overhead perspective. The axes of rotational symmetry for the electrodynamic trapping field and gravity pass through the viewing plane. The droplet is depicted at a considerably larger size for visibility.**

A typical experiment involves the stable levitation of a ~100 μm diameter charged droplet in the EDB. The temperature of the
trap is set to the desired value at least an hour before performing experiments. In order to maintain a humid environment and
reduce droplet evaporation, at the start of each experiment, a water volume is placed into a cavity in the bottom electrode to
serve as a source of water vapour. Using this method, a relative humidity ranging from 60% to 85% under conditions of steady
state was achieved within the EDB. The relative humidity in the EDB was estimated by measuring the droplet evaporation rate
and applying a steady state evaporation model (eq. 13-21 in Pruppacher and Klett, 1997). During the experiment, the light
scattered by the oscillating droplet is recorded while sweeping the frequency $f_e = 1/2\pi \cdot \omega_e$ over a predetermined range. In
our experiments, the frequency sweep is carried out in two separate ranges, namely 9 kHz to 17 kHz and 21 kHz to 45 kHz,
within a duration of either 5 or 10 seconds. In the range between 17 kHz and 21 kHz the HF transformers employed in our
generator possess their own resonance with strongly varying output amplitudes. Therefore, this frequency range was avoided.



In the future, an improved generator design should allow for a continuous tuning of the excitation frequency. During the sweep,
each frequency value is held constant for a duration of 100 ms before moving to the next frequency value. The amplitude of
the applied excitation voltage is kept constant during the frequency sweep. Angular resolved light scattering images are
recorded every 1 s to determine the droplet size. Further frequency sweeps can subsequently be applied to follow the temporal
evolution of $\eta$ and $\sigma$ if desired.

It is important to note that the PMT electronics constitute a low-pass filter to the signal. This introduces an additional
frequency-dependent phase shift in the experimental data. To account for this, the following equation is utilized:

$$\phi_c = \phi_S - tan^{-1}\frac{f_e}{BWL}, \tag{8}$$

where $\phi_c$ is the corrected phase, $\phi_S$ is the phase shift obtained by symmetry analysis, $f_e$ is the frequency of the excitation
signal and $BWL$ is the bandwidth limit of the PMT (200 kHz). For excitation frequencies ranging from 9 to 45 kHz, the phase
shift correction resulting from the PMT bandwidth limit ranges from 2.6° to 12.7° illustrating the substantial impact of the
correction.

Droplets of nanopore water (thermo scientific, GenPure Pro UV) and of dilute aqueous sucrose solutions with concentration
of 1 wt% and 2.5 wt% were investigated. The solutions were prepared by dissolving sucrose (VWR, proteomic grade, >99.9 %)
in water. We studied water as it is a fundamental constituent of the atmosphere and its viscosity and surface tension are well
known while sucrose is used as a proxy for organic aerosol and features a strongly concentration dependent viscosity.
Measurements were taken at temperatures between ~0°C and -31°C with at least 10 to 15 droplets at each temperature.

### 2.3 Phase analysis method

The light scattering intensity from a droplet that undergoes quadrupole deformations according to Eq. (4) is a complex function
of the deformation coordinate $\alpha$ and eludes a direct inversion. To determine surface tension and viscosity it is necessary to
measure the phase shift between droplet response and applied excitation. This can be achieved by focusing on the extrema of
droplet deformation and their phase shift relative to the maxima of the excitation. Around the turning points of the droplet
deformation, the light scattering function should be symmetrical in time. This is explored in the "cycle symmetry method"
which we introduce in this work and which is explained in detail in the SI. In brief, we calculate a scalar symmetry parameter
for each point in time for the recorded light scattering data. The maxima of this symmetry time series, i.e. the moments in time
of highest local symmetry, correspond to the turning points of the droplet oscillation. It turns out that the cycle symmetry
method is a very robust method of determining the phase shift between droplet oscillation and excitation. The method can
easily be applied to data taken with a changing excitation frequency as is the case during a frequency sweep as long as the
frequency changes slowly compared to the vibrational period. The result of such an analysis of a frequency sweep on a levitated
water droplet of R = 43.2 μm and at a temperature of T = 0.4°C is shown in Figure 2. A fit to the data according to Eq. (5) is
shown as a red line. The fit yields values for $\gamma$ and $\omega_0$. Viscosity and surface tension are then obtained using Eq. (6) and (7).



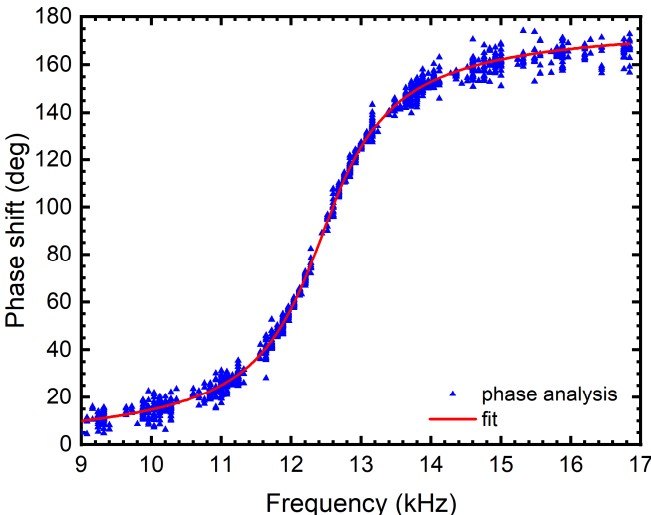


**Figure 2. Phase-frequency data obtained from the symmetry analysis method for a water droplet with R = 43.2 µm at T = 0.4°C. Also plotted is a fit according to Eq. (5).**

### 3 Results and discussion

The phase shift of induced droplet oscillations was measured to determine the viscosity and surface tension of individual water

droplets and individual sucrose solution droplets from -1°C to -31°C. The ensemble average of the viscosity for ten water droplets at each measurement temperature is shown in Figure 3. The error bars shown in the plot represent the uncertainty associated with the fit and the uncertainties from independent radius and charge determination. The largest contribution to the uncertainty is the determination of the droplet radius. Overall, the results agree well with literature values. However, at temperatures below -20°C, we observe a deviation of the measured average values compared to the literature values. These

low temperature values also have larger uncertainties which could be attributed to a weaker light scattering signal arising from smaller amplitude oscillations as a result of increasing viscosity at lower temperatures.



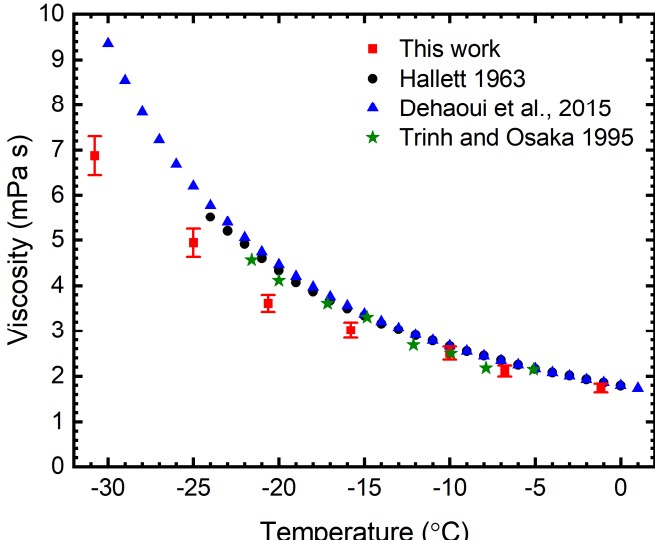

**Figure 3. Experimentally determined viscosity values of water with respect to temperature. The red data points are the average viscosity for ten levitated water droplets and the error bars represent the relative error in the fit. Literature values are also shown**
**for comparison.**

The literature values for the viscosity of supercooled water shown in Figure 3 were determined using a range of different methods. Bulk measurements were performed by Hallett (1963) who used a capillary flow technique to measure the viscosity of water at temperatures down to -23.8°C. Whereas Dehaoui et al. (2015) studied Brownian motion of polystyrene spheres suspended in water to determine the viscosity of supercooled water down to -34°C. They indicate that the uncertainties

associated with their data range from 2.3 % at the highest temperature to 2.8 % at the lowest temperature. A drawback to their method is that they require a reference value of viscosity at a known temperature, meaning their method is not widely applicable to a range of compounds. Furthermore, polystyrene spheres are often pre-coated with surface active molecules to prevent coagulation and this coating might affect their mobility and hence the viscosity measurement. The most relevant literature values to this study are those determined by Trinh and Ohsaka (1995) who performed measurements on individually

acoustically levitated supercooled droplets. They determined the decay time following induced shape oscillations in acoustically levitated spheres to determine the viscosity down to -21°C.

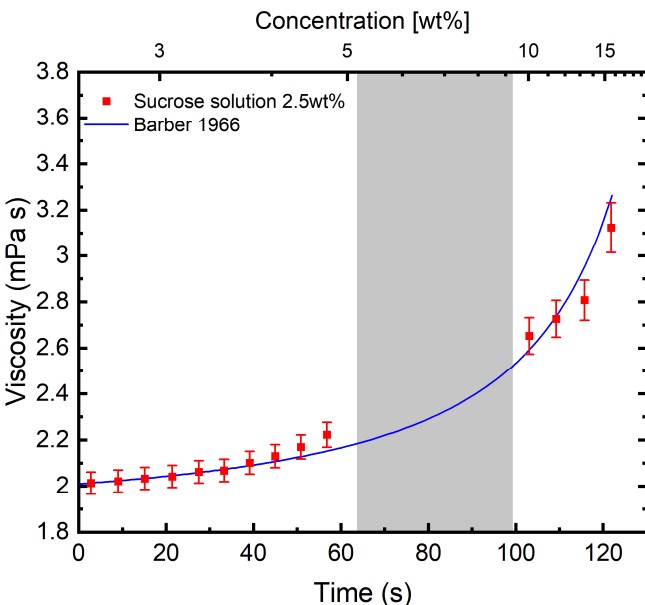

**Figure 4. Viscosity of an evaporating sucrose solution droplet with initial concentration of 2.5 wt% as a function of evaporation time at T = -1.5°C. The line represents calculated values according to a parameterization for the viscosity of sucrose solution as function**
**of concentration and temperature (Barber, 1966). The grey shaded area between ~64 and 100s indicates the time range in which the natural frequency of the droplet oscillation was between the two frequency sweep ranges. Here, the fitting procedure did not yield reliable results which explains the gap in the data.**

The measured viscosity values for an evaporating sucrose solution droplet of initial concentration 2.5 wt% are shown in Figure 4. Using the measured size evolution of the droplet and knowledge of the initial concentration at droplet injection, the

concentration as function of time can be calculated. The relationship between concentration and time was used to calculate the upper axis. We note, that the data is in excellent agreement with the parameterization published by Barber (1966). In this work, the data was analysed after completion of the experiment. However, it is worth noting that in principle, the analysis can be performed while the droplet is still levitated in the EDB and that changes in viscosity and surface tension could be followed in in situ.

In addition to the viscosity, we calculated the surface tension using Eq. (7) and the experimentally determined values of radius at the natural frequency, mass and charge of the droplet. The resulting ensemble average of the surface tension for ten droplets each at every temperature setting is shown in Figure 5. Error bars represent the uncertainty derived from the fit, the radius measurement and the charge measurement as the standard deviation of the ensemble average was typically smaller than the uncertainties derived from the independent variables.





Comparison with the literature values in Figure 5 shows that the surface tension of water is a weak function of temperature. The values measured in the current study are lower than the literature values and possess more scatter. The different literature measurements agree well with each other and the majority were determined using a technique based on the capillary rise method (Hrubý et al., 2014; Kalová and Mareš, 2021; Vinš et al., 2015) apart from Trinh and Ohsaka (1995) who determined surface tension by determination of the natural frequency for the fundamental mode of shape oscillations.

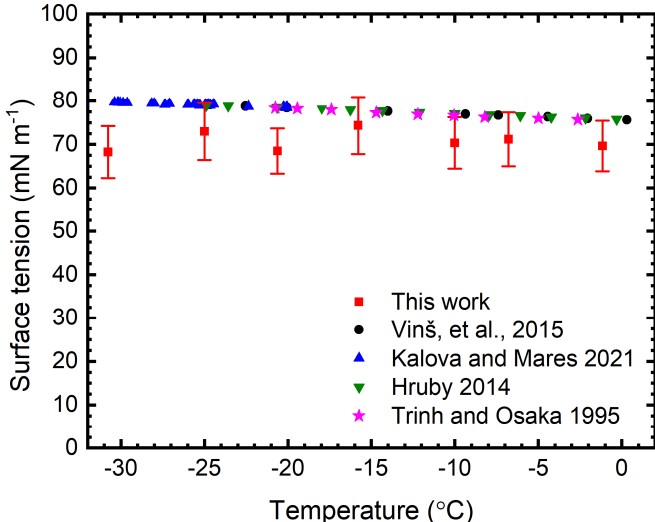


**Figure 5. Experimentally determined surface tension of supercooled levitated water droplets and comparison to literature values. Red data points correspond to the average surface tension value for ten water droplets and error bars represent the relative error in the fit.**

It is well known that levitated droplets can readily absorb potentially surface active substances from the surrounding air in the
EDB affecting surface tension measurements (Bzdek et al., 2015; Duft et al., 2002). Similarly, it is possible that a small amount of surfactant present in the sample solution can concentrate on the surface during evaporation of the droplet. We investigated this effect by analysing the surface tension of individual droplets as they evaporated in the EDB, as shown in Figure 6. Here, one pure water droplet, two droplets of 1 wt% and two droplets of 2.5 wt% initial sucrose concentration were observed over time. Panel (a) of Figure 6 shows the surface tension plotted as a function of droplet radius as the droplets evaporate with time.

As the droplets evaporate, their surface tension decreases indicating a possible accumulation of surfactants on the droplet surface. As in the viscosity analysis, the measured radius change is used to calculate the concentration in each droplet as function of time, resulting in a plot of surface tension as function of concentration (Fig.6b). To the best of our knowledge no data on the surface tension of sucrose solution below 20°C is available. Nevertheless, to be able to put our results into perspective we estimated the surface tension of sucrose solution at low temperatures using data measured at 25°C from a study




performed by Aumann et al. (2010). We then used the well-known data for water to scale the sucrose solution values according to $\sigma_s(\text{T}[°C]) = \sigma_w(\text{T}[°C])/\sigma_w(25°C) \cdot \sigma_s(25°C)$ to lower temperature. The resulting reference curve is plotted as a solid line in Figure 6b. The comparison of our data with the reference clearly shows, that the observed decrease in surface tension as the droplets evaporate is not due to the increase in sucrose concentration. However, the similarity of the decrease when plotted against the radius may indicate an increase in surfactant concentration and may hence explain the decreasing surface tension.

A second observation in Figure 6b is the much more pronounced difference in surface tension at initial sucrose concentration. We plan to further investigate these effects quantitatively in a forthcoming paper.

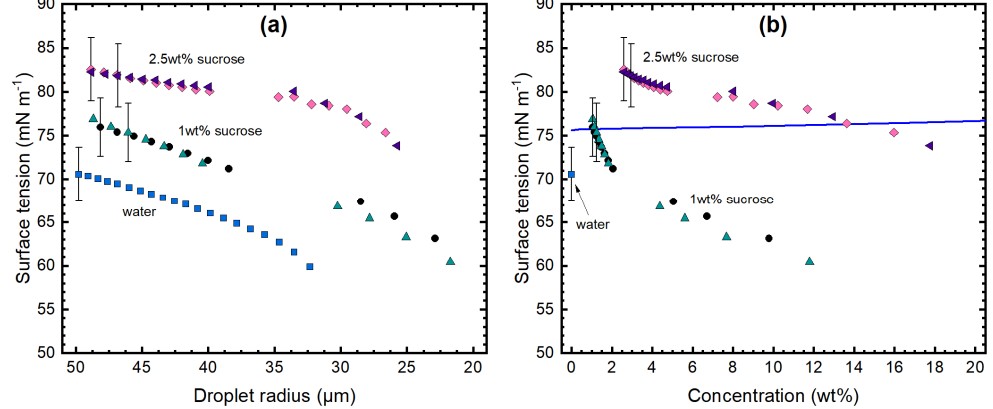

**Figure 6. Surface tension of water and sucrose solution droplets. Shown are the results from five individual droplets at a temperature of T = 0.2°C. Two droplets at two initial sucrose concentrations (1 wt% and 2.5 wt%) and one droplet of pure water. Panel (a) shows**
**the evolution of the surface tension as the droplets evaporate in the EDB. The presence of gaps in the data series indicates that the natural frequency of the droplet has shifted beyond the resonance of the excitation transformer preventing analysis of data in that range. Panel (b) depicts the surface tension plotted as function of the concentration as calculated from the initial sucrose concentration and droplet radius under the assumption that only water evaporates from the droplet. For water, only the initial data point is plotted. The solid line represents the expected dependency for the surface tension derived from literature as detailed in the**
**text. To enhance clarity, error bars are displayed solely for one data point in each series in both panels.**

    The novel method reported here is directly applicable to the determination of viscosity and surface tension at atmospherically relevant supercooled temperatures and can also be applied at ambient temperature. As previously mentioned, the majority of contactless methods used to determine aerosol viscosity and surface tension have predominantly focused on measurements at ambient temperature and above (e.g. Athanasiadis et al., 2016; Bzdek et al., 2015; Endo et al., 2018; Fitzgerald et al., 2016;
Power and Reid, 2014; Power et al., 2013; Richards et al., 2020; Tong et al., 2022) and tend to focus on concentrated droplets. A key advantage to the method reported here is that it utilizes only one analysis method; the symmetry analysis of droplet shape oscillations, to determine both droplet viscosity and surface tension over a wide temperature range (ambient to supercooled). Other studies involving coalescence of two optically levitated droplets use multiple analysis methods to cover different viscosity ranges and determine surface tension (e.g. Bzdek et al., 2020; Power and Reid, 2014). Furthermore, the



study by Trinh and Ohsaka (1995) who induced shape oscillations in acoustically levitated droplets used two different methods to determine surface tension and viscosity; analysis of resonance frequency of oscillations and relaxation time to a sphere respectively. Having one analysis method to determine both viscosity and surface tension as described in the present study has additional advantages in terms of providing insight into the phase state of complex mixtures e.g. for water droplets coated in an organic film, information on the viscosity of the water droplet and the surface tension of the shell could be deduced from a

single experiment. Additionally, the method described here allows the viscosity and surface tension to be calculated independently using two separate equations.

Regarding surface tension, the technique allows a measurement to be made within about 2s after droplet levitation allowing only little time for contaminants to accumulate on the droplet surface. This is a shorter timescale than that obtained with the contactless method that interrogates the resulting oscillations of two coalesced optically trapped droplets 10s after they are

individually trapped (Bzdek et al., 2015).

Furthermore, the current study only requires small-scale oscillations (amplitude of tens of nm) meaning that more atmospherically relevant smaller droplets can be probed, in comparison to larger scale oscillations of 1-2 % in diameter for a mm sized acoustically levitated droplet (Trinh and Ohsaka, 1995).

In comparison to non-levitation techniques such as the study by Dehaoui et al. (2015), our method does not require the

knowledge of any other material property apart from density to derive the viscosity similar to Hallett (1963).

To date, there remain few measurements of viscosity and surface tension for atmospherically relevant organic aerosols and organic containing mixtures at supercooled temperatures. As aerosols uptake water and grow to become cloud droplets their concentration will likely be diluted. It is therefore important to determine viscosity and surface tension of dilute substances as well as concentrated ones. The viscosity range covered by the current proof of concept measurements ranges from 2 to 7 mPa s.

The results indicate the potential for the new technique to explore dilute organic solutions at supercooled atmospherically relevant temperatures. Further studies are required to determine the full extent of the range of viscosities using this method, however, we hypothesize that the technique could be used to measure viscosity values within some part of the semi-solid range ($10^2$-$10^{12}$ Pa s, Koop et al. (2011)) for droplets containing dilute organics.

As demonstrated by the surface tension measurements in Fig. 7, we are able to measure small changes in viscosity and surface

tension in real time. Therefore, the technique is ideally suited to investigate the role of atmospheric processing on the viscosity and surface tension of solution droplets in equilibrium with surrounding water vapour such as relative humidity, photochemistry and oxidation.

## 4 Conclusion

We have developed a new technique to determine viscosity and surface tension of levitated droplets at supercooled

temperatures. A high frequency AC signal swept over a variable frequency range is used to excite shape oscillations in a levitated droplet. From the analysis of light scattered by the droplet as it undergoes shape oscillations we are able to determine



the phase shift of oscillations compared to the excitation signal which can then be used to determine droplet viscosity and surface tension. The proof-of-concept measurement has been demonstrated for supercooled water droplets and sucrose solution droplets over a temperature range of ~0°C to –31°C. The technique represents a step forward in the simultaneous measurement
of viscosity and surface tension of levitated droplets at supercooled temperatures utilising only one analysis method.

*Code availability.* The code for the phase analysis is available from the authors upon request.

*Data availability.* The data are available from the authors upon request.

*Author contributions.* The project was conceptualized by TL. The methodology was developed by MS, SHJ, DD, and TL. The HF generator was designed by TL and implemented by MS. The experiments were carried out by MS and SHJ. The phase analysis code and various software tools were developed by DD. The collected data was analysed by MS, SHJ, and DD. The initial draft of the manuscript was prepared by MS, SHJ, and DD. The manuscript was reviewed and edited by MS, SHJ, DD,
with contributions from AK and TL. Throughout the project, supervision was provided by AK, DD, and TL.

*Competing interests.* The authors declare that they have no conflict of interest.

*Financial support.* SHJ was supported by a DFG grant under the grant number JO 1849/1-1. MS acknowledges funding of a
research grant for postdocs by the Alexander-von-Humboldt Stiftung. TL, AK, DD acknowledge financial support by the Helmholtz Association under Atmosphere and Climate Program (ATMO). The article processing charges were covered by the Open Access publication fund of the Karlsruhe Institute of Technology (KIT).

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
