# Peer review of "The viscosity and surface tension of supercooled levitated droplets determined by excitation of shape oscillations"

_EGUsphere, 2023_

## Author Response (AR1)

The authors would like to thank the editor and the editorial staff for handling the manuscript and express their gratitude to the referees for dedicating time to reviewing the manuscript and providing concise comments and questions. We have thoroughly addressed each of the points raised and indicated changes made in the manuscript by highlighting them in the text. Below are the referee's comments in black text followed by our replies in blue text followed by the specific changes in the revised manuscript in orange text.

**RC1:**

1. What is the driver voltage amplitude and how is the appropriate magnitude of this chosen?

The magnitude of the applied AC-voltage was kept below 1.5kV peak-to-peak in order to make sure that no electrical discharges occur within the EDB. Due to the application of two independent AC-potentials, i.e. the levitation and the excitation potentials, potential differences equivalent to the sum of both amplitudes will occur. The operational amplifier was programmed in such a way as to compensate the nonlinear output of the transformers. In this way it was possible to maintain a constant output amplitude across the entire frequency range.

We added this information to the manuscript text on page 7 as well as to the SI in section 2:

"In our experiments, we applied frequency sweeps across two distinct ranges: 9 kHz to 17 kHz, and 21 kHz to 45 kHz, while maintaining a constant excitation amplitude of approximately 1.5 kV peak-to-peak. The duration of individual frequency sweeps was selected as either 5 or 10 seconds."

2. What is the uncertainty in the particle size, and how much of this translates to uncertainty in viscosity and surface tension? Equations 6 and 7 both show dependencies on R^2 or higher, indicating some significant propagation of error. The authors chose to use the phase function approach, which for large particles can yield reasonable accuracy when comparing measured spectra to Mie theory simulations. There are several approaches to sizing using these data, as described in the SI. Thus, it would be convenient to describe how the angular scattering pattern is used in the main text along with the estimated uncertainty.

As outlined in Section 3 of the SI, the relative uncertainty in particle size is contingent upon whether single images (4% uncertainty) or a filtered set of multiple images (1% uncertainty) were utilized for analysis, with the resulting R(t) data being effectively smoothed through the application of an evaporation model. The uncertainty in droplet size significantly contributes to the overall uncertainty in viscosity and surface tension and improving the size determination method would provide great benefit to a future iteration of the method.

Because of its significance, we added a short description of the size determination method to the main text on pages 6-7, while keeping the detailed description in Sect. 3 of the SI:

"Images of the angular resolved light scattering (phase function) are recorded every 1 s utilising a CCD camera. From these images the droplet size is obtained by analysing the spatial frequency of fringes in the phase function and by comparison with the frequency from calculated phase functions using Mie theory. A more detailed description is provided in Sect. 3 of the SI."

3. I would suspect that when the droplet is driven at a frequency resonant with a mode of oscillation that the amplitude of the oscillation would increase significantly, leading to a higher scattered signal from the droplet. Using a dual-phase lock-in, the amplitude of the scattering

and the phase of the signal, both as a function of frequency, could be analyzed as the driver frequency is swept. Has amplitude information been measured and/or analyzed to identify the resonant frequency? This would presumably directly yield omega_0 for an underdamped oscillation, but perhaps not yield sufficient information on the damping constant for viscosity analysis.

We determined $\omega_0$ and the damping constant $\gamma$ by analysing the oscillation amplitude. However, we found that amplitude analysis was associated with a significantly greater uncertainty compared to phase analysis and that it is only reasonably effective for non-evaporating or extremely slowly evaporating droplets (dr/dt<<10nm/sweep). The reason for this lies in the highly nonlinear characteristics of the measurement signal, specifically in how the intensity of scattered light varies with the oscillation coordinate.

To obtain absolute values for the amplitude of oscillation (i.e. amplitude of actual droplet deformation), it would be necessary to develop an additional analysis algorithm. For the purpose of testing, we manually compared measured and simulated light scattering time series to obtain droplet deformation amplitudes. The corresponding result has been included in panel B of Fig S11 in the SI. However, this demanded a considerable investment of manual analysis effort, which we found to be unjustified when contrasted with the straightforward and relatively accurate results achieved through phase analysis.

These factors led us to focus on the phase analysis for our methodology.

4. What are the limits of viscosity that can be probed with this method. The viscosity values reported here are all many orders of magnitude lower than what is typically considered to be a viscous particle. Is there a limit on the viscosity that can be measured due to the limited response of a more viscous particle to any kind of shape deformation? Is this connected with the deviation of the measure viscosity in this approach from the literature range (Figure 3 for example)?

In our approach, we employ a phase-sensitive detection (PSD) technique. Fundamentally, PSD proves to be highly effective in extracting minute signals concealed amidst significant background noise levels. As a result, the amplitude of the signal and the extent of the oscillation that can be induced are unlikely to pose concerns using this methodology.

However, there might be other factors limiting the viscosity range in our method, such as obtaining a sufficient amount of data to precisely determine the natural frequency of the undamped oscillation $\omega_0$ and the damping constant $\gamma$ from the theoretical fitting process of the phase-frequency data. Ideally, this entails carefully selecting the frequency sweep range, encompassing $\omega_0$ where the phase shift reaches $\pi/2$, and encompassing a sufficiently broad span of both lower and higher frequencies. This is particularly effective for underdamped oscillations. However, for overdamped oscillations, the amplitude-frequency curve and the phase-frequency curve adopt a relatively flat profile, which indicates a diminished information embedded within these curves. Consequently, the accurate extraction of $\omega_0$ and $\gamma$ using our methodology can become challenging under such circumstances. Based on this reasoning, we estimate an upper limit for the viscosity measurement in the range of a few Pa s, while the determination of a lower limit remains elusive. The detailed limits of viscosity will be explored in further measurements.

We have added this statement to the main text on page 13:

"We estimate an upper limit for the viscosity measurement of a few Pa s for the current experimental set-up (see Sect. 6 of the SI), while the determination of a lower limit remains elusive. However, further studies are required to determine the full extent of the range of viscosities using this method."

5.  Were measurements performed on particles that spanned a range of charge states? When calculating the charge according to the SI, no gas flow drag factor is reported. How does the gas flow in the chamber affect the force balance? How big is the uncertainty in the charge and how much does this contribute to the calculation of surface tension and viscosity?

The span of droplet charge states measured in this study was between 0.67 pC and 0.84 pC. We estimated the relative uncertainty in droplet charge measurement to be about 3.5%. We added this information to the SI on page 5 and a statement in the main text referring to the SI on page 9. A systematic study of the influence of droplet charge on this method was not performed.

Equation (6) in the manuscript indicates that the viscosity is independent of the droplet charge. However, following Eq. (7) the surface tension is dependent on Q. To evaluate a potential influence of the charge on this method it is convenient to rewrite Eq. (7) as:

$$1 = \frac{\rho \omega_0^2 R^3}{4\sigma} + X,$$

where $X = Q^2/Q_R^2$, the fissility parameter, with droplet charge $Q$ and Rayleigh charge $Q_R$. From Eq. (7) in the above form it becomes apparent, that the contribution due to the term containing the droplet charge is small, as long as the fissility is much smaller than 1. For the droplets under investigation in this study, the fissility is typically below X=0.01 upon injection and never exceeds X=0.1 throughout the measurement duration. Consequently, the uncertainty in the charge does not contribute significantly to the uncertainty in the surface tension. Instead, it is the power dependence on the droplet radius, that emerges as the primary contributing factor.

In addition to this analysis, the uncertainties of dependent variables are consistently calculated following the principles of error propagation.

Regarding the gas flow, the EDB was operated without any gas flow, eliminating the need to account for drag force in the force balance. Gas flows through the EDB have been utilised in previous experiments in our group.

In order to avoid ambiguity with these experiments, we added the above information to the main text on page 5:

"The EDB is operated without gas flow during individual experiments, eliminating the need to account for a drag force acting on the levitated droplet."

6.  What is the timescale for evaporation in the measurements shown in Figure 6? Does the decrease in surface tension arise from the adsorption of know contaminants from an external source (i.e. are there lubricant oils that might be outgassing, low purity gas cylinders, etc.), or are these present in the particle and become more concentrated at the surface as evaporation occurs?

Figure 6 in the manuscript shows surface tension measurements for five individual droplets (1 water and 4 sucrose solution droplets) while evaporating in the EDB. Each of these measurements was

recorded over a time span ranging from 100 to 120s. We added this information to the main text on page 12:

"Here, one pure water droplet, two droplets of 1 wt% and two droplets of 2.5 wt% initial sucrose concentration were observed over a time span ranging from 100 to 120 s."

The decrease in surface tension could arise from adsorption of contaminants or those that are already present in the solutions studied. However, it is difficult to say for certain without further investigation. Special care was not taken to prepare solutions in an ultra-clean environment. As stated in the response to comment 5, no gas flows were used during the experiments. However, in the event that discharges occurred, the EDB was purged with laboratory grade synthetic air which is prepared on site from liquidised nitrogen and oxygen. Low purity gas cylinders and lubricant oils were not utilized. As in previous experiments in our group involving EDBs great care was taken to keep the EDB in a clean condition.

7. How large are the oscillations in the particle shape?

Obtaining the oscillation amplitude from the measurement is challenging due to the complex nature of the time dependent light scattering signal. See also our reply to comment 3. Nevertheless, by manual analysis of the signal we found that oscillation amplitudes as small as a few Angstroms could be analysed.

This is further supported by theoretical evaluation of Eq. (1) from the manuscript. The theory shows that the oscillation amplitude of the equatorial radius of a water droplet at 0°C (R=50μm, Q=0.8pC) reaches a value of 1.5nm off-resonance (5kHz, 1kVpp) and up to 12nm at the resonance (11kHz, 1kVpp).

To provide the reader with an understanding of the estimated range of the oscillation amplitude, we have included a statement in the main text that presents our estimate on page 8:

"In analogy to the phase shift, the amplitude of the oscillation could in theory also be analysed in order to determine viscosity and surface tension. For the example shown in Fig. 2 the oscillation amplitude in droplet radius was between 3.5 Å off-resonance (17 kHz), and 5 nm on-resonance (12.5 kHz). We would like to highlight that oscillations in droplet radius as small as a single monolayer of water can be detected. However, due to the complex nature of the light scattering signal as a function of the oscillation coordinate, we found amplitude analysis to be more error prone compared to the phase shift analysis."

8. What size range of particles can be explored using this technique? Being able to access particles that span a wide range of surface-volume ratios would facilitate an exploration of surface partitioning and the influence of surfactant depletion etc.

We estimate that for our experimental setup the range in droplet diameter is limited between 5 to 200μm. Please refer to our reply to comment CC1 for further information. We added a statement on the size limitations of this setup to the manuscript on page 6:

"We estimate, that the EDB allows the stable levitation of droplets between 5 and 200 μm in diameter."

For this technique it may turn out to be challenging to overcome these limits whilst simultaneously utilizing electrodynamic levitation and electrodynamic excitation. However, the combination of

acoustic or optical levitation with electrodynamic excitation could potentially open up the possibility of accessing a broader range of droplet sizes.

**RC2:**

1. The only comment I would like to ask the authors to consider is that they explain/estimate in more detail the limitations of the technique in both upper viscosity values and range of surface tension are.

We now elaborate in more detail on the limitations of the technique with respect to the range in viscosity and surface tension. We have included the following on pages 13-14:

"We estimate an upper limit for the viscosity measurement of a few Pa s for the current experimental set-up (see Sect. 6 of the SI), while the determination of a lower limit remains elusive. However, further studies are required to determine the full extent of the range of viscosities using this method. We hypothesize that the technique could be used to measure viscosity values for droplets containing dilute organics. Regarding surface tension, we estimate that the method is applicable within a range of at least $10^{-3}$ to 1 N/m, thus encompassing the complete spectrum of values typically encountered for atmospheric aerosols."

And in the conclusion on page 14:

"We estimate that the method is suitable for viscosities below a few Pa s and surface tensions falling between $10^{-3}$ and 1 N/m."

2. I was particularly interested in understanding the upper limit for the viscosity. In order to gain a better feeling for the behavior of the phase shift, I put some numbers into eq. (7) to estimate the natural angular frequency, $\omega 0$, namely (R=50 μm, σ=70 mN/m, ρ=1000 kg/m3, Q= 0.8 pC) and came up with 47 kHz. Using eq. (6) with a viscosity of 2 mPa s, I calculate a γ of roughly 4000. Putting those numbers into eq. (5), I cannot reproduce something similar to what is shown in Fig. 2. Most likely this is a mistake on my side, but the authors could provide in the SI some numbers on, $\omega 0$ and provide a plot where they show the expected phase shift assuming log spaced viscosity data keeping all other parameters constant.

We cannot say for certain what caused the discrepancy between this calculation and our results. In order to facilitate the comparison, we included the droplet charge, as well as results from the fitting procedure in the caption of Fig. 2.

Revised Fig. 2. Caption page 8:

"Figure 1. Phase-frequency data obtained from the symmetry analysis method for a water droplet with R = 44.3 μm and Q=0.82 pC at T = 0.4°C. Also plotted is a fit according to Eq. (5) with fit parameters $\gamma = 4639\ s^{-1}$ and $\omega_0 = 2\pi \cdot 12.464\ kHz$. Application of Eq. (6) and (7) yielded the following values for viscosity, $\eta = 1.82\ mPa\ s$ and surface tension, $\sigma = 68\ mN\ m^{-1}$."

We have added an additional section to the SI, Section 6, where we show the expected phase shift as well as the amplitude of oscillation for a range of droplet viscosities. We anticipate that this will help in reproducing our results.

**Additional section 6 in SI:**

**"6. High viscosity limit**

To illustrate a potential limitation of the method at higher viscosity we computed the phase shift according to Eq. (5) from the manuscript for various values of the viscosity ranging from 2 mPa s to 1 Pa s. The result is shown in panel A of Fig. S11 for a droplet of R=50 μm, ρ=1000 kg/m³, Q=0.8 pC, σ=0.07 N/m. For viscosity above a few Pa s the phase frequency curve becomes rather flat indicating that a high uncertainty in the fitted parameters can be expected. To provide an idea about the width of the resonance and the amplitude of oscillation, we also calculated the corresponding oscillation amplitude. The result is shown in panel B of Fig. S11. Here, all calculated curves are scaled in the vertical direction by an identical factor which was determined by matching the 2mPas curve to data retrieved from a manual analysis of a single water droplet measurement at about 0 °C (~2 mPa s). The graph shows that the resonance peak disappears for viscosity above about 20 mPas and that the amplitude reduces even further posing an additional challenge for detecting the oscillations at higher viscosity.

[Figure]

***Figure S11*** *Calculated phase shift and amplitude of oscillation for a water droplet as function of excitation frequency in the EDB for various viscosity values. Droplet parameters are given in the text.*

Based on these considerations, we can conclude that a viscosity above a few Pa s represents a current upper limit for this method."

Initiated by the referee's comment we also checked our equations and calculations. While doing so we noticed a typo in eq. (5) in the manuscript which might be contributing to the above discrepancy. We corrected the equation in the revised manuscript on page 5:

$$\tan \phi_e = \frac{2\gamma \, \omega_e}{\omega_0^2 - \omega_e^2}$$

The typo does not impact the results presented in the manuscript as in our analysis the correct equation was utilized.

Furthermore, the value R=43.2μm originally cited in the manuscript as the droplet radius in the text above and in the caption of Fig. 2 does not actually correspond to the droplet radius. Instead, it corresponds to the time in seconds when the sweep transverses $\omega_0$. The accurate droplet radius at this moment was R=44.3μm according to $R^2(t) = R_0^2 - b \cdot t$ with $R_0 = 50 \, \mu m$ and $b = 12.48 \, \mu m^2 s^{-1}$ as given in the caption to Fig. S3 of the SI and using $t = 43.2 \, s$. Incidentally, both numbers are very similar and the incorrect number was inadvertently entered into the manuscript. We have replaced the number with the correct droplet radius.

Regarding the last mentioned point of the referee's comment, $\omega_0$ is the frequency of oscillation of the undamped oscillator. This parameter is therefore by definition independent of viscosity. It is a noteworthy and interesting feature of the damped driven oscillator, that the frequency of the undamped system still appears in the equations of the damped system. The position of 90° in the phase-frequency curve which indicates $\omega = \omega_0$ consequently is invariant to the viscosity of the droplet. This property might be considered as an advantage of analysing the phase instead of the amplitude response for some mildly overdamped systems.

3. Fig. S4: Please explain what is causing the apparent decrease in droplet charge after 70 s in a bit more detail. What is the size of the particle at this time of evaporation? Is this reaching the stability limit of the EDB or is the DC-feedback loop to keep the droplet in the center of the EDB no longer working?

The size of the droplet can be taken from Fig. S3 as both data series were recorded from the same levitated droplet. We included this information in the figure caption of Fig. S4 of the SI:

"**Figure S4** Calculated droplet charge as function of time of the levitated water droplet shown in Fig. S3."

Deviations in the charge data like the one presented in Fig. S4 are usually not connected to the stability limit of the EDB as both occur independently. In the example shown in Figures S3 and S4, the time series ends at 160s because the droplet reached the stability limit for the chosen fixed EDB settings. However, the deviation in the charge curve occurred significantly before.

To explore the origin of the deviation in the charge curve is outside the scope of this paper. However, we will provide a more detailed explanation below which hopefully sheds some more light on this.

The charge curve shown in Fig. S4 was calculated using Equation E1 which in turn was derived under the assumption that the droplet resides in the electrodynamic centre of the EDB, i.e. at z=r=0. This is the singular location in the EDB where the electric force due to the AC levitation potential (the pseudopotential) vanishes and only gravity and static DC electric force act on the droplet. Outside the EDB centre the average electric force due to the AC-levitation field has to be taken into account in the force balance. If the droplet is not positioned in the EDB centre, application of equation E1 results in deviations from the constant charge as visible in Fig. S4. Based on the graph, we make an estimation that places the droplet's centre of mass approximately 5µm above the EDB centre. This assessment relies on treating the droplet's charge as a point charge, as opposed to a surface charge. Consequently, we recommend to exercise caution with this estimation.

4. Connected to the data shown in Fig. S4: How is the flow in the EDB affecting the determination of Q based on the applied DC-field? Does the drag force cause a systematic uncertainty here?

The EDB was operated without airflow during our experiments. Airflow was employed only for purging purposes between experiments. We included a statement in the manuscript to avoid ambiguity with previous studies made in our group where airflows have been utilized simultaneous to droplets being levitated. The following statement was inserted on page 5:

"The EDB is operated without gas flow during individual experiments, eliminating the need to account for a drag force acting on the levitated droplet."

We note, that if the flow conditions are well known, the drag force can be included in the force balance. Much like in Millikan's famous experiment, the air flow does not hinder determining the droplet charge as long as the flow conditions or the relative velocity of the particle in the air are well known.

**Further changes made to the manuscript:**

Page 1, line 14; Page 13, line 337

Based on our now refined assessment of the range limits of the viscosity we removed the inclusion of "within the semi-solid range" from the abstract and the discussion.

Page 3, line 95

We removed the sentence: "Currently, viscosity values for dilute droplets are derived from models." This sentence was a remnant from a previous version of the manuscript among the co-authors, and we believe it is not relevant anymore in the context of this paragraph.

Page 5, line 152:

We added information on the ambient conditions surrounding the levitated droplet. "Charged droplets are levitated in a quadrupole EDB enclosing a 1cm³ volume of air at ambient pressure."

Page 6, line 176 – page 7, line 180

We added a statement concerning the consideration of the evaporative cooling effect on the droplet temperature. "The temperature of the levitated droplet is offset with respect to the ambient temperature in the EDB due to the evaporative cooling effect. The relative humidity in the EDB, as well as the temperature offset due to evaporative cooling is estimated by measuring the droplet evaporation rate and applying a steady state evaporation model (eq. 13-21 in Pruppacher and Klett, 1997). The droplet temperature is always determined by adding the evaporative cooling offset (between -1.2 and -1.8 °C) to the ambient temperature in the EDB."

**Further changes made to the SI:**

- A typo was corrected in equation E1
- A typo was corrected in the EDB geometry constant stated on page 5 below equation E1